# Substrate DNA Promoting Binding of *Mycobacterium tuberculosis* MtrA by Facilitating Dimerization and Interpretation of Affinity by Minor Groove Width

**DOI:** 10.3390/microorganisms11102505

**Published:** 2023-10-07

**Authors:** Aadil Ahmed Memon, Xiang Fu, Xiao-Yong Fan, Lingyun Xu, Jihua Xiao, Mueed Ur Rahman, Xiaoqi Yang, Yu-Feng Yao, Zixin Deng, Wei Ma

**Affiliations:** 1State Key Laboratory of Microbial Metabolism, School of Life Sciences and Biotechnology, Shanghai Jiao Tong University, 800 Dongchuan Road, Shanghai 200240, China; 2Shanghai Institute of Infectious Diseases and Biosecurity, Shanghai Public Health Clinical Center, Fudan University, Shanghai 200032, China; 3Shanghai Huaxin Biotechnology Co., Ltd., Room 604, Building 1, Tongji Chuangyuan, No. 99 South Changjiang Road, Baoshan District, Shanghai 200441, China; 4Laboratory of Bacterial Pathogenesis, Institutes of Medical Sciences, School of Medicine, Shanghai Jiao Tong University, Shanghai 200025, China

**Keywords:** *Mycobacterium tuberculosis*, MtrA, quantitative affinity, minor groove width (MGW), dimerization

## Abstract

In order to deepen the understanding of the role and regulation mechanisms of prokaryotic global transcription regulators in complex processes, including virulence, the associations between the affinity and binding sequences of *Mycobacterium tuberculosis* MtrA have been explored extensively. Analysis of MtrA 294 diversified 26 bp binding sequences revealed that the sequence similarity of fragments was not simply associated with affinity. The unique variation patterns of GC content and periodical and sequential fluctuation of affinity contribution curves were observed along the sequence in this study. Furthermore, docking analysis demonstrated that the structure of the dimer MtrA-DNA (high affinity) was generally consistent with other OmpR family members, while Arg 219 and Gly 220 of the wing domain interacted with the minor groove. The results of the binding box replacement experiment proved that box 2 was essential for binding, which implied the differential roles of the two boxes in the binding process. Furthermore, the results of the substitution of the nucleotide at the 20th and/or 21st positions indicated that the affinity was negatively associated with the value of minor groove width precisely at the 21st position. The dimerization of the unphosphorylated MtrA facilitated by a low-affinity DNA fragment was observed for the first time. However, the proportion of the dimer was associated with the affinity of substrate DNA, which further suggested that the affinity was actually one characteristic of the stability of dimers. Based on the finding of 17 inter-molecule hydrogen bonds identified in the interface of the MtrA dimer, including 8 symmetric complementary ones in the conserved α4-β5-α5 face, we propose that hydrogen bonds should be considered just as important as salt bridges and the hydrophobic patch in the dimerization. Our comprehensive study on a large number of binding fragments with quantitative affinity values provided new insight into the molecular mechanism of dimerization, binding specificity and affinity determination of MtrA and clues for solving the puzzle of how global transcription factors regulate a large quantity of target genes.

## 1. Introduction

Mycobacterium tuberculosis, the most lethal pathogen, caused 10.6 million new incidences and 1.6 million deaths in the year 2021 alone [1]. New research directions are urgently required to address this global challenge [2,3,4].

Bacterial pathogens initiate and modulate multifarious processes and functions that are essential for infection, such as metabolism, growth and signaling, to adapt to the host environment and prevail over host immune systems [5]. For instance, to survive inside human macrophage cells, M. tuberculosis must first mimic or modify the host signaling pathways and cellular functions by expressing eleven eukaryotic-like serine/threonine protein kinases (PknA to PknL) and two serine/threonine phosphatases (PtpA and PtpB) [6]. Then, bacilli can survive and even grow by utilizing lipids as their sole carbon source [7] and expressing gene coding for isocitrate lyases to generate energy through the glyoxylate cycle [8], along with a series of fatty acid importers [9,10], hypoxia [11] and genes for the synthesis of lipolytic enzymes, such as virulence-associated lipids, polyketide, poly-acylated trehaloses, sulfolipids and mycolic acids. In addition, dormancy [12] and antibiotic tolerance [13] genes must also be functional [8]. The above complex processes demonstrate the crucial role of global virulent transcriptional regulators (Global TF) for the bacterial pathogen during infection. In accordance with expectations, the virulence of *M. tuberculosis* was significantly attenuated [14] when PhoP, which regulates more than two hundred genes [15], including ESAT-6 [16], sigK, atpA-H, rplS and lepB, was knocked out or knocked down in vivo [17].

MtrA, the only essential response regulator of M. tuberculosis [18,19], belongs to the winged helix-turn-helix (wHTH) sub-class and the highly diverse OmpR subfamily [20,21], which are commonly present in viruses, bacteria and some eukaryotes, including humans [22]. The functions of around 270 MtrA target genes [23,24] reported so far were involved in cell wall biogenesis, lipid, coenzyme transport and metabolism, energy production, protein turnover, modification and secondary metabolite biosynthesis [24,25,26,27,28,29,30], which suggests the complex process and mechanism of MtrA binding to the promoter regions of target genes in addition to their important functions.

Several studies on protein–DNA complex structure indicated that prokaryotic wHTH sub-class members bound to their target DNA as a tandem dimer [17,31,32]. Most of the studies suggested that in the OmpR family, the α4-β5-α5 face is responsible for the formation of the dimer [33,34] that is stabilized by an extensive network of salt bridges comprised of charged residues like Lys, Glu, Asp, Arg and Asp in α4, β5 and α5 to surround the hydrophobic patch formed by Ile, Leu, Val and Ala in α4 and α5 [17,33,35].

However, one to four hydrogen bonds were also reported in the α4-β5-α5 face of ArcA [33], KdpE [31] and TorR [35], and one was usually formed between the two DNA-binding domains (DBD) [32].

After TFs were phosphorylated in vitro before co-crystallization, several studies concluded that phosphorylation on the RECs of OmpR family members was essential for dimerization [36,37]. However, phosphorylation could considerably increase its binding affinity to target DNA [5,33], and acetylation could completely inactivate its binding activity [19].

The α-helix-8 of DBD recognizes the target DNA by inserting into a major groove and forming a hydrogen bonds network between the side chains of the participating residues and the nucleotide bases [38], and 6–7 bps direct repeats separated by 4 bps spacer were frequently identified in the binding motifs [17,31,39]. However, for this sub-class, the binding sequences were highly diverse, and the binding domains were not characteristically conserved at both sequence and structural levels [40].

Nevertheless, hydrogen bonds formed between the PhoP residue of R237 [17] and PmrA T208 [32] in the C-terminal β hairpin known as “wing” [22] and the base, and the sugar and phosphate backbone [41] of the minor groove of target DNA [42] was associated with sequence recognition. For example, the wing domain of human RFX1 was reported to recognize substrate DNA through binding to the minor groove [43], but the wing of heat shock TFs (HSF) significantly improved the DNA binding specificity through modulating dimerization rather than directly contacting with the DNA [40].

Though the shape readout [44] theory has become increasingly popular as it reveals the mechanism of binding specificity [45] and the importance of minor groove width (MGW) [46,47,48], the studies identifying DNA shape features mainly rely on high-throughput sequencing with poor repeatability and expensive and time-consuming computational approaches [45]. Presently, due to the lack of an efficient, cost-effective research system to generate sufficient quantitative affinity data, owing to the high sequence diversity of most TF binding sites [49], the association between sequence and binding affinity cannot be completely represented by any single model built so far, including the k-mer-based model, PWM-based models and DNA shape [50].

This study attempts to explore the mechanism of MtrA specificity and affinity determination comprehensively from sequence to structure so as to elucidate how MtrA differentially regulates hundreds of target genes.

## 2. Materials and Methods

### 2.1. Chemicals and Reagents

The chemicals used in this study were purchased from Sigma Aldrich (St. Louis, MO, USA).

### 2.2. Bacterial Strains and Growth Conditions

Bacterial strains: *M. tuberculosis* H37RV, *E. coli* TOP10 and BL21(DE3) were sourced from the stocks of our laboratory.

Bacterial culture conditions: *E. coli* was grown in LB medium at 37 °C 220RPM. *M. tuberculosis* H37Rv was cultured in Middlebrook 7H9 broth and plated in the 7H10 [51], and 50 μg/mL of kanamycin was added when needed.

### 2.3. Cloning, Expression and Purification of MtrA

A fragment encoding MtrA *(Rv3246c)* was amplified from the *M. tuberculosis* H37Rv genomic DNA using Phanta Max Super-Fidelity DNA Polymerase (Vazyme, Nanjing, China). The primers (5′-cagcaaatgggtcgcggatccATGGACACCATGAGGCAAAGG-3′), (5′-gtggtggtggtggtgctcgagTCACGGAGGTCCGGCCTT-3′) were designed using Primer3 (https://www.primer3plus.com, accessed on 23 November 2020) and synthesized by Tsingke Biotechnology (Beijing, China). The purified fragment was cloned into pET28a and linearized by *BamH* I and *Xho* I (NEB, England) double-digestion using Exnase (Vazyme, Nanjing, China).

*E. coli* BL21 (DE3) transformed with pET28a-MtrA was induced with 0.5 mM IPTG at 16 °C for 20 h when OD600 reached 0.6. For protein purification following routine procedures, a 250 mM final concentration of Imidazole was used for elution after being loaded on a Ni-NTA 6FF prepacking chromatographic column (Sangon Biotech, Shanghai, China), and eluted MtrA was desalted and concentrated using a 10 Kda protein concentrator (Millipore, MA, USA) in storage buffer (20 mM HEPES, 100 mM NaCl and 5% glycerol pH 7.5). The purified protein was stored at −80 °C for further experiments after the purity and concentrations were assayed by 12% SDS-PAGE (Yeasen Biotechnology, Shanghai, China) and a BCA kit (Sangon Biotech, Shanghai, China).

### 2.4. Binding Affinity Assay

The binding affinity of the MtrA to fragments was assayed using a ForteBio Octet RED 96 and an NTA sensor (ForteBio, Fremont, CA, USA), which are principally bio-layer interferometry assay instruments for quantifying the interactions between macromolecules [52]. Experiments were generally conducted as described in [53] with some modifications. Oligonucleotide fragments used were synthesized by Tsingke Biotechnology (Beijing, China). A total of 100 nM proteins and 500 nM dsDNA and buffer (20 mM HEPES, 20 mM NaCl, 50 mM CaCl_2_, 10 mM MgCl_2_, 10 mM KCl, pH 7.5) were used for the test. The assay program was set up at 30 °C; first baseline for 2 min; protein loading for 5 min; DNA association for 5 min; dissociation for 5 min, followed by a 5 s regeneration. Negative (no binding) DNA was used as a reference. A 1:1 binding model was applied to globally fit the binding isotherms, and kinetic parameters such as kon, koff, KD and R^2^ were analyzed using Octet Analysis Studio Software (Version 7.0).

### 2.5. Calculation of Position-Specific Affinity Contribution

The One-Hot program was used to convert the base of each position in each sequence into a four-dimensional vector in the order “ATCG”. Here, M number of sequences with a length of N bp was converted into a matrix *B* ∈ R^(M × 4N). The matrix of *D* ∈ ℝ^*M*×1^ was established after the affinities of sequences were processed with a logarithm (*log*_10_). Then, the correlation matrix *E* ∈ R^(1 × 4N) between each column of matrix *B* and *D* was calculated using the Pearson correlation coefficient algorithm implemented by the Python Pandas library. Finally, the correlation matrix *F* ∈ R^(N × 4) was achieved by reshaping matrix *E*.

### 2.6. Size Exclusion Chromatography

The samples of protein DNA mixture were analyzed by AKTA Pure 25 M (GE, Boston, MA, USA) using EzLoad 16/60 Chromdex 200 pg (Bestchrom, Shanghai, China) at 25 °C and buffer (20 mM HEPES, pH 7.5, 200 mM NaCl, 50 mM CaCl_2_, 10 mM MgCl_2_ and 10 mM KCl) flowing at 0.2 mL/min after MtrA (10 µM) associated with dsDNA fragments (20 µM) in the same buffer and incubated for 30 min at 25 °C. The sequences of the DNA fragment used for size exclusion chromatography were synthesized by Tsingke Biotechnology (Beijing, China) and are listed in Table 1.

### 2.7. Phylogenetic Tree Analysis

The phylogenetic tree of MtrA binding sequences was constructed using the neighbor-joining algorithm in MEGA X [54] with 1000-bootstrap sampling. The accession numbers of the sequences used for alignment for α4-β5-α5 of OmpR subfamily are listed in Table 2.

### 2.8. Molecular Docking

The dimer structure of MtrA was predicted using SWISSMODEL (https://swissmodel.expasy.org, accesed on 30 September 2022) utilizing the dimer crystal structure of KdpE (4KFC) as the template based on the monomer structure of MtrA (2GWR) and the quality of the model was evaluated by a Ramachandran plot (http://www.ebi.ac.uk, accessed on 2 October 2022) The structures of PhoP (5ED4) and PmrA (4S04) were used for structural alignment with MtrA.

HDOCK (http://hdock.phys.hust.edu.cn/, accessed on 5 October 2022) was used for protein–DNA docking using the default settings, where MtrA and the B-DNA sequence were defined as the receptor and the ligand, respectively, and docking results were analyzed and illustrated using PyMOL (Version 2.5).

### 2.9. Interface Residue Analysis

PDBePISA (https://www.ebi.ac.uk/pdbe/pisa/, accessed on 8 October 2022) was used for the protein–protein and protein–DNA interface analysis.

### 2.10. DNA Shape Analysis

The shape of the DNA fragment was analyzed using DNAshape (https://rohslab.usc.edu/DNAshape, accessed on 15 october 2022); the data obtained were processed and analyzed using OriginPro 2019.

## 3. Results

### 3.1. Unique Features of MtrA Diverse Binding Sequences

The binding affinities of MtrA to its 294 binding fragments ranged widely from 4.62 × 10^−8^ M(Rv2524c) to 9.88 × 10^−3^ M (Rv3246c), spanning five orders of magnitude (Table 3). The affinity values of all 294 binding fragments are listed in Appendix A.

The sequence conservation analysis indicated that the 5th–11th and 16th–22nd regions were relatively conserved regardless of affinity (Figure 1). Among the relatively conserved positions, like 6–7th, 9th, 11th, 17–18th and 20–21st, the proportion of G at the 17th and 6th positions and T/C at the 21st position remained constant, while that of 5th A/T, 7th T, 9th A, 11th C, 16th C/T, 18th T and 20th A declined with a decrease in affinity.

Among the eight sub-clusters grouped by sequence phylogenetic analysis, the highest-affinity sequences (KD < 1 × 10^−8^ M) Rv2524c and Rv1918c were found in sub-clusters 1 and 8, while of the 11 higher-affinity sequences (KD < 1 × 10^−7^ M), one was located in sub-cluster 7; two in sub-clusters 3, 4 and 8, respectively; and four in sub-cluster 5 (Figure 2a). The affinity values of sequences sharing high sequence similarity were not similar; for example, in sub-cluster 8, the affinity of Rv1918c (7.90 × 10^−8^ M) was 1000-fold higher than that of Rv3849 (9.65 × 10^−3^ M) (Figure 2b).

The GC content at the 6th, 11th and 17th positions (97.62%) was above 80%, while that of the 7th (3.40%), 9th, 18th and 20th positions was lower than 20%, and the GC content fluctuated uniquely and drastically in both regions of the 6th–11th and 17th–22nd bp (Figure 3).

To explore the role of the conserved regions with special GC variation patterns, both regions of Rv2524c were replaced with the same region with the confirmed non-binding sequence (CGGGACG). The affinity decreased 84,800-fold when the 6th–11th bp region was replaced, and no further binding was detected when the 16th–22nd region was replaced (Table 4).

### 3.2. Position-Specific Affinity Contributions of MtrA sequences

The binding affinity contributions of each nucleotide for each position were calculated by analyzing the association of every position for 294 sequences with their affinity values by statistical analysis. As shown in Figure 4, a significant difference in affinity contribution was identified among 26 positions, from the highest value of 15.7% (8th, T) to the lowest of 1.8% (6th, G). T contribution values were higher than that of the other bases at 14 positions, while A, G and C were responsible for three, six and three positions, respectively, and the dominance of T became more obvious as, in the top five contributing positions (7th T (10.3%), 8th T (15.7%), 15th T (11.0%), 18th T (13.4%) and 25th G (10.3%)), four positions were T except for the 25th position (G). Each kind of base’s contribution curve fluctuated periodically along the sequence; meanwhile, the four curves wavered sequentially in major regions. Interestingly, in all 294, no A was observed at the 10th position, and no C was observed at the 6th and 17th positions. Contrary to expectations, the affinity of the artificial sequence comprised of the highest contribution bases at all positions (5′-GTTTTGTTATCGTTTTGTGATATGGA-3′) (2.48 × 10^−7^ M, R^2^ 0.956) was 3.97-fold lower than the natively highest affinity, Rv2524c.

### 3.3. Substrate DNA Facilitating the Dimerization of MtrA

A unique peak was detected at 91.07 mL when MtrA was analyzed solely by gel filtration chromatography (Figure 5a), while one main peak (87.03% area) and a small peak (12.97% area) were detected at 76.28 mL and 89.15 mL, respectively, after MtrA was incubated with a high-affinity fragment (Rv2524c) (Figure 5b). The proportion of the main peak at 76.23 mL decreased to 72.72%, and that of the second peak at 90.69 mL increased to 27.28% (Figure 5c) when the medium-affinity fragment (Rv3859c, 1.11 × 10^−5^ M) was present. Moreover, the main peak shifted to the position of 84.73 mL (93.74% area), leaving one small peak at 74.26 mL (6.29% area) when the low-affinity (Rv0309, 1.78 × 10^−4^ M) fragment was present (Figure 5d). Surprisingly, the peak was detected at 89.89 mL rather than 91.07 mL when it was confirmed that no binding fragment was present (Figure 5e).

### 3.4. Prediction of MtrA Dimer–DNA Complex

#### 3.4.1. Prediction of MtrA Dimer Structure by Homology Modeling

Since the similarity between MtrA and KdpE was higher at both sequence (38.94%) and structural (48%) levels than that of the other homolog proteins with structural data (Figure 6), the dimer structure of MtrA was predicted by homology modeling using the KdpE dimer (4kfc) as a template (Figure 7a).

The evaluation of the established MtrA dimer model showed that 88.4% of residues were in the most favored regions (A, B, L), 9.7% were observed in the additional allowed regions (a, b, l, p), 1.6% were observed in the generously allowed regions (~a, ~b, ~l, ~p) and only one residue (0.3%) was found in the disallowed regions (Figure 7b).

#### 3.4.2. MtrA Dimer Interface Analysis

Among the 17 hydrogen bonds identified by interface analysis, 15 hydrogen bonds were formed between RECs, but only 2 were formed between DBD. The six Arg residues were involved in nine hydrogen bonds (Arg115, Arg117, Arg119, Arg121, Arg122 and Arg170), and the residue Glu96 was involved in three hydrogen bonds (Table 5).

Additionally, Arg170, Val172 and Glu129 were only observed in MtrA_1_, while Leu95, Ala99, Asp139, Ala142 and Asn123 were identified in MtrA_2_ only. Furthermore, eight hydrogen bonds formed between Arg117(αH5)-Glu96(αH4), Arg119(αH5)-Asp100(Coil-9), Arg115(αH5)-Asp101(β-5) and Arg115(αH5)-Tyr102(β-5) were found within the α4-β5-α5 face, which comprised the hydrogen bond network with a symmetric complimentary structure (Table 5 and Figure 8).

The sequence alignment of mycobacterium homologous genes of MtrA, PhoP and PhoB from representative species revealed that the αe4-β5-α5 regions were conserved among G+ and G− bacteria, and each gene had its own unique characteristics. For example, in MtrA, the residue at the 101st position was Val (hydrophobic) rather than Arg or Lys (positively charged) for the rest, and the 106th residue of PhoB was Thr (polar) instead of Glu (negatively changed) (Figure 9).

### 3.5. MtrA Dimer–DNA Complex Structure Analysis

The docking analysis of homo-dimer MtrA with Rv2524c demonstrated that the binding domains of two subunits, α-Helix-8 (191TRLVNVHVQRLRAKV205), inserted into the major groove of the 5th–11th and 16th–22nd regions correspondingly, and the residues Arg219 and Gly220 of the wing (218VRGV221) interacted with the minor groove in front of the binding box (docking score: −231.29; confidence score: 0.7800) (Figure 10).

### 3.6. The MGW Was Affected by Nucleotides at Its Own and Flanking Position, and MGW at the 21st Position Significantly Influenced the Affinity

When the 20th G of Rv3859c was replaced with A, the binding affinity was significantly increased by 73.7-fold; meanwhile, DNA shape analysis revealed that the value of MGW at the 21st position decreased from 4.80 Å to 4.63 Å. The affinity was considerably decreased by 534.3- and 328.3-fold when the 20th G was substituted with C or T, respectively, and the corresponding MGWs were 5.05 Å and 4.84 Å, respectively. When the 21st T of Rv3859c was further replaced with G or A as well as G20A, though their values of MGW at the 21st position were 4.67 Å and 4.46 Å, the corresponding affinities were decreased by 18.0% and 46.6%, respectively, compared with the wild type and 113.37-fold lower than that of G20A, and their MGWs at the 22nd position (4.53 Å and 4.33 Å) were narrower than that of the 21st position correspondingly. Nevertheless, the affinity of G20A and T21C increased 106.3 times; its MGW at position 21 was 4.45 Å. In short, the substitution of the base affected not only the MGW of its own position but also that of positions up to 3 bps both upstream and downstream (Table 6 and Figure 11).

## 4. Discussion

The specific binding of TFs to their targeted DNA sequences is the crucial step in which pathogenic bacteria modulate multifarious processes, from metabolism and growth to virulence, through gene expression regulation. Thus far, the studies underlying the mechanism of TF binding affinities were eukaryote-dominated and mainly based on the semi-quantified sequence-affinity data of different TF families derived from the Chip-Seq [55]. However, much fewer data are available in terms of prokaryotic TF binding affinity and specificity [56]. In this study, the mechanism of MtrA’s binding and affinity determination was explored structurally through a comprehensive investigation of its 294 highly diverse binding sequences with a wide range of quantitative affinities.

No clear affinity-associated motif or nucleotide at a specific position was identified in this study (Figure 1). Even in the distribution of high-affinity sequences (KD < 1 × 10^−7^ M) in the phylogenetic tree, the differential affinity values of fragments shared high sequence similarity (Figure 2); furthermore, the unique and drastic fluctuation of the GC content and the periodical and sequential waves of the affinity contribution curves suggested that the binding affinity was not simply determined or even associated with nucleotides at specific positions or motifs, like binding boxes.

The fact that nucleotide substitution at the 20th and/or 21st positions of Rv3859c resulted in a change in affinity of four orders of magnitude revealed the importance of these two positions in MtrA binding and combination of flanking nucleotides in affinity determination. The above findings are consistent with the report that the effect of specific nucleotides on the affinity of the Forkhead (FOX) TF family could be rescued by flanking nucleotides [56]. The strong negative association between the value of MGW precisely at the 21st position and the affinity (Figure 11) proved by DNA shape analysis of nucleotide substitution at the 20th and/or 21st positions further provided quantitative proof of the DNA shape readout theory [47,48,57].

The MtrA binding motif, supported by the MtrA dimer–DNA complex model, GC-content and affinity contribution curves (Figure 3, Figure 4 and Figure 9), and the box replacement experiment (Table 4) was generally consistent with other OmpR family members resolved by crystallography, such as length of box and spacer [58].

The gel filtration chromatography analysis showed that unphosphorylated MtrA was completely monomer before binding with DNA, and the proportion of dimer was significantly associated with the affinity of substrate DNA (Figure 5), which proved that substrate DNA can facilitate the dimerization of unphosphorylated MtrA regardless of affinity. Moreover, the above findings imply that the affinity can actually be considered as one characteristic of the stability of dimers.

Thus far, studies on the TF-DNA complex structures of the OmpR subfamily solved by co-crystallization after protein phosphorylation in vitro have concluded that the dimerization of KdpE [30], PhoP [17] and PmrA [31] was facilitated by phosphorylation on REC [59,60].

However, it was found that another OmpR subfamily member, TorR, could bind as a dimer to its high-affinity recognition sequences in vitro [35,61] without phosphorylation. We found that unphosphorylated MtrA could bind to its substrate DNA in vitro, regardless of affinity. These findings provide more convincing experimental proof to support Barbieri, C. M. et al.’s hypothesis that the dimerization of the OmpR family response regulators could be facilitated by the binding of DNA in addition to phosphorylation [59].

Among the 15 hydrogen bonds in the inter-REC domains, 14 were formed within α4-β5-α5 and involved its nine residues; furthermore, 8 hydrogen bonds were formed between the α4-β5-α5 faces of two MtrA molecules in a symmetrical and complimentary way. The above findings demonstrate the core position of α5 in this network and imply that hydrogen bonds might play an equally important role in salt bridges and hydrophobic patches [33] in dimerization, at least for MtrA. Additionally, the high sequence conservation of α4-β5-α5 throughout Gram-positive and -negative bacteria and the unique conservation pattern of specific genes, like K/R101V in MtrA and E106T in PhoB, further indicated its important roles in dimer formation and stabilization (Figure 9). The interesting findings of this study provide insight into the mechanism of global TF regulating a large number of genes and molecular mechanisms of binding specificity and affinity determination. New research strategies and techniques, such as artificial intelligence, should be introduced into studies on the complex binding mechanisms of prokaryotic TFs, along with genetic manipulation, structure studies, etc.

## Figures and Tables

**Figure 1 microorganisms-11-02505-f001:**
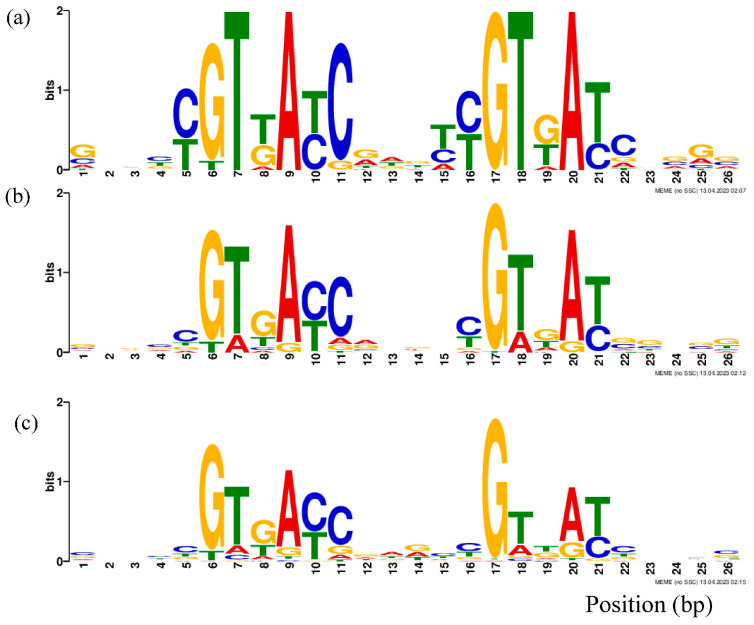
The sequence conservation of different affinity groups of MtrA: (**a**) high-affinity (1 × 10^−7~−8^ M) group; (**b**) medium-affinity group (1 × 10^−5~−6^ M); (**c**) low (1 × 10^−3~−4^ M)-affinity group.

**Figure 2 microorganisms-11-02505-f002:**
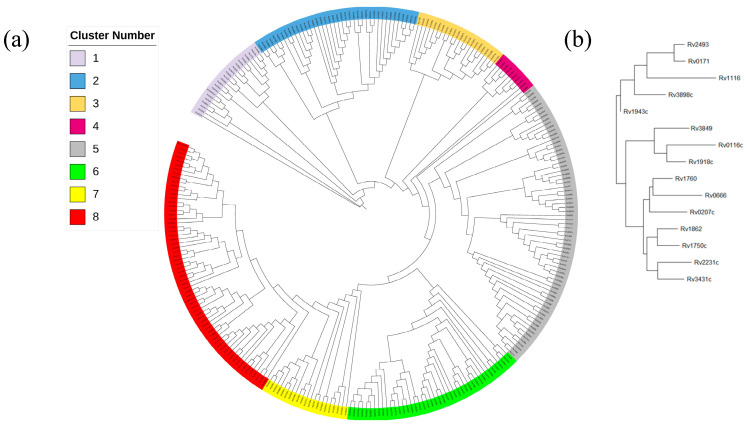
Phylogenetic analysis of MtrA binding sequences. (**a**) Complete tree; (**b**) sub-cluster 8.

**Figure 3 microorganisms-11-02505-f003:**
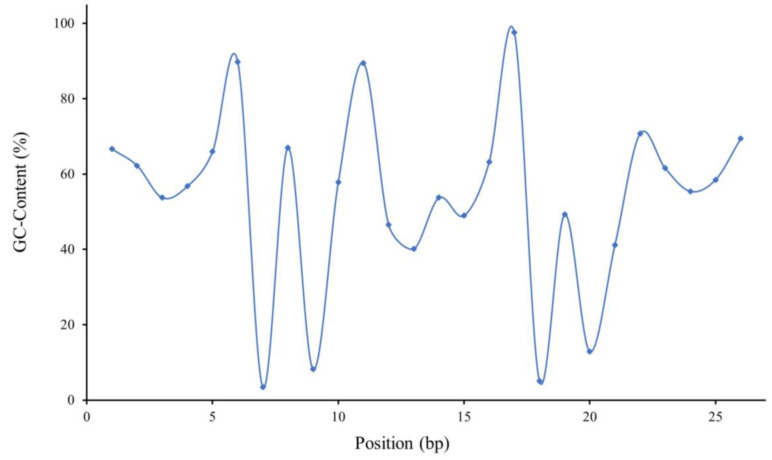
Position-specific GC content analysis of MtrA binding sequences.

**Figure 4 microorganisms-11-02505-f004:**
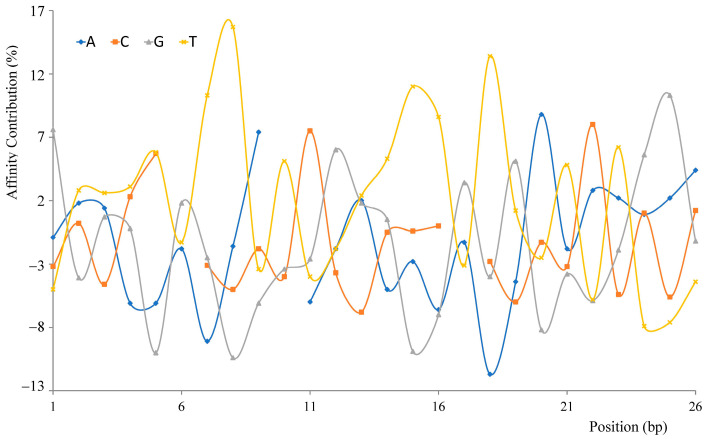
Affinity contribution of specific base at each position of binding sequences.

**Figure 5 microorganisms-11-02505-f005:**
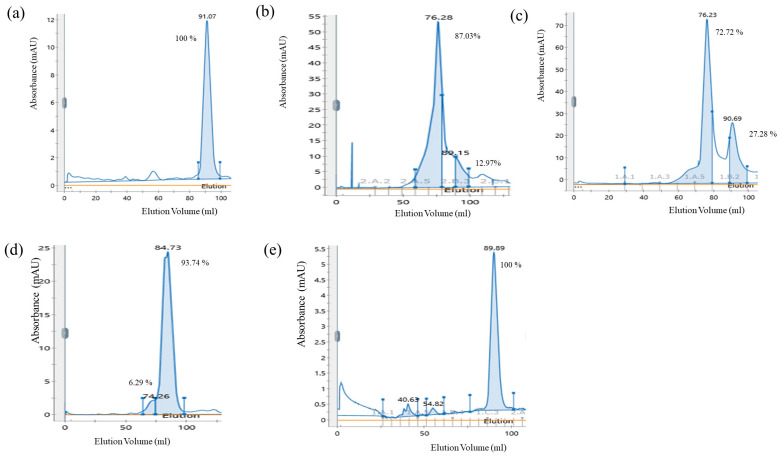
Interaction of MtrA with DNA was analyzed by gel filtration chromatography: (**a**) MtrA only; (**b**) with Rv2524c (high affinity); (**c**) with Rv3859c (medium affinity); (**d**) with Rv0309 (low affinity); (**e**) with no binding fragment.

**Figure 6 microorganisms-11-02505-f006:**
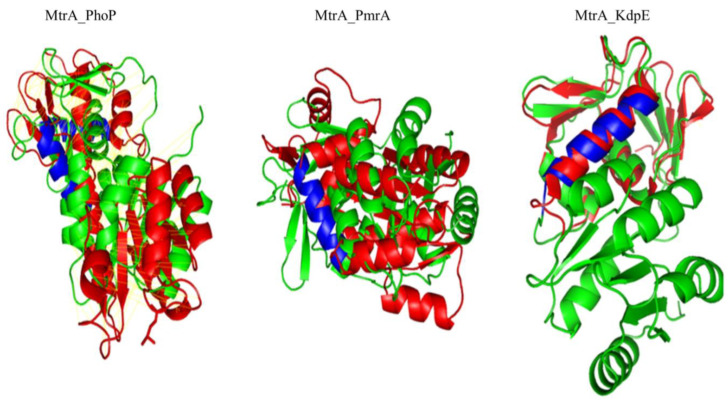
Structural alignment of OmpR family protein: Red—MtrA; green—PhoP, PmrA, KdpE; blue—α-helix 8 of DBD.

**Figure 7 microorganisms-11-02505-f007:**
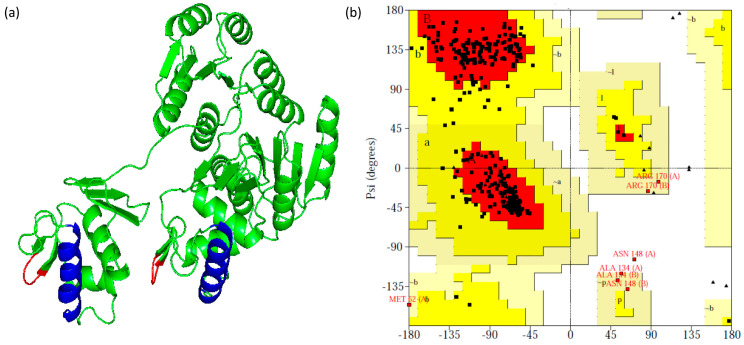
Structure prediction of MtrA dimer: (**a**) Cartoon model: blue—α8; red—wing. (**b**) Result of Ramachandran plot: red—most favored regions; yellow—additional allowed regions; light yellow—generously allowed regions; white—disallowed regions.

**Figure 8 microorganisms-11-02505-f008:**
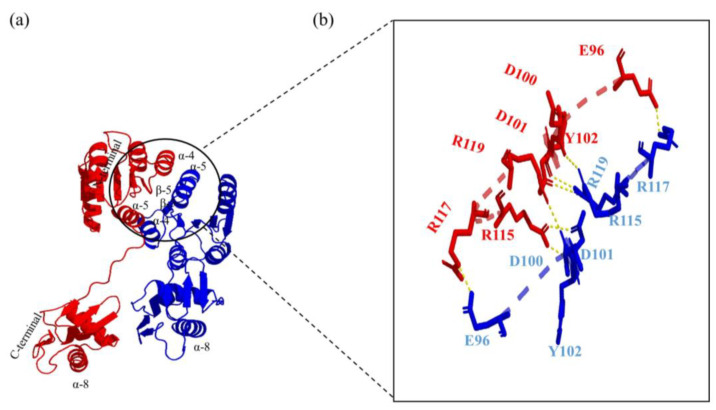
Interface of MtrA dimer and hydrogen bonds within α4-β5-α5 face: (**a**) cartoon model of the MtA dimer; (**b**) hydrogen bond network in which two MtrA molecules were colored with red and blue.

**Figure 9 microorganisms-11-02505-f009:**
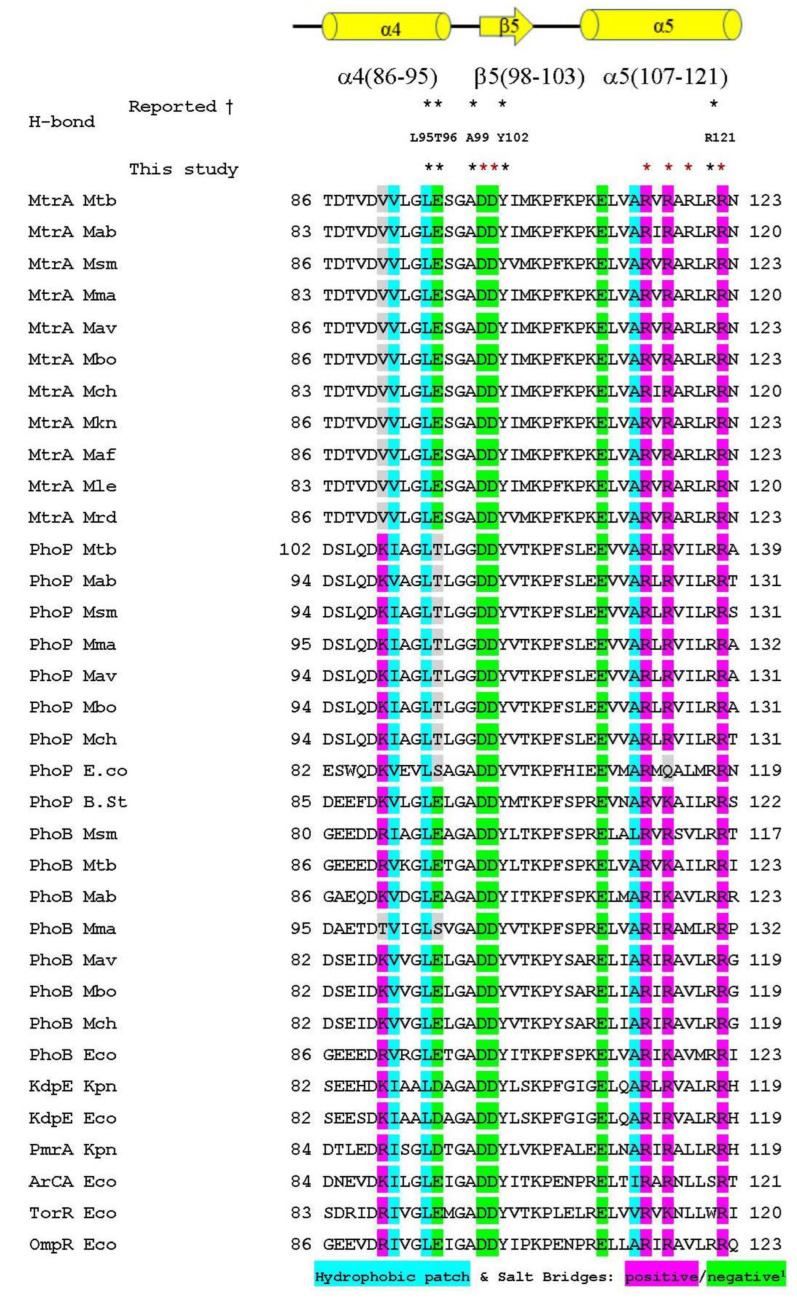
Comparison of α4-β5-α5 from selected mycobacterium OmpR subfamily. †: Numbering follows MtrA; *: the residues involved in forming hydrogen bonds, * in red were identified in this study, * in black were reported ones. ^1^: Hydrophobic, salt bridegs and charged risidues were refered to the report of Toro-Roman et al. [33]. All the abbreviations are as follows: Mtb—*M. tuberculosis*; Mab—*M. abscessus*; Msm—*M. smegmatis*; Mma—*M. marinum*; Mav—*M. avium*; Mbo—*M. bovis*; Mch—*M. chelonae*; Mkn—*M. kansasii*; Maf—*M. africanum*; Mle—*M. leprae*; Mrd—*M. rhodesiae*; E.co—*Escherichia coli*; Kpn—*Klebsiella pneumonia*; B.st—*Bacillus subtilis*. The accession numbers of sequences used in the figure are listed in Table 2, and the rest were from reference [32].

**Figure 10 microorganisms-11-02505-f010:**
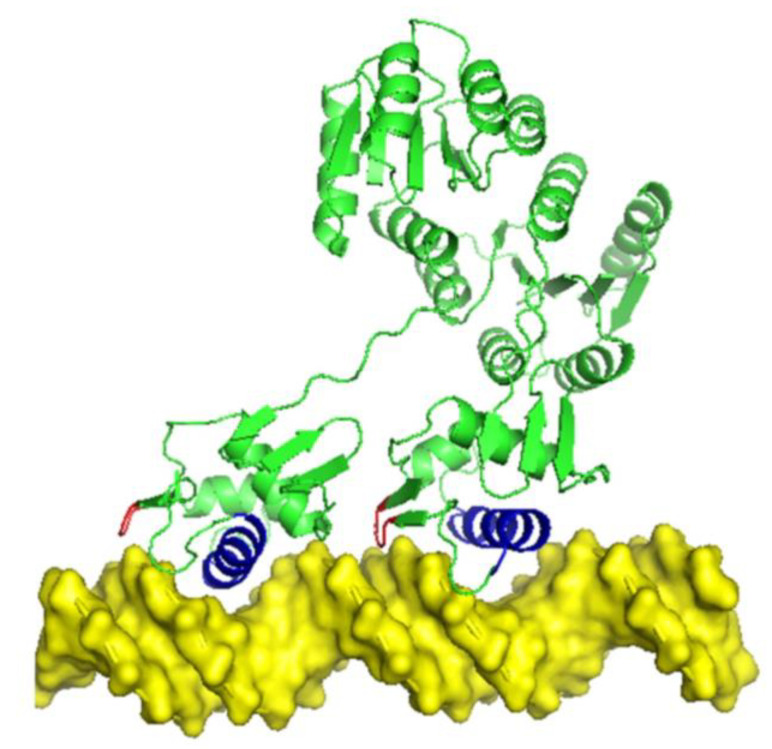
Overview of dimer MtrA-Rv2524c complex structure. Note: the domains are colored as Figure 7.

**Figure 11 microorganisms-11-02505-f011:**
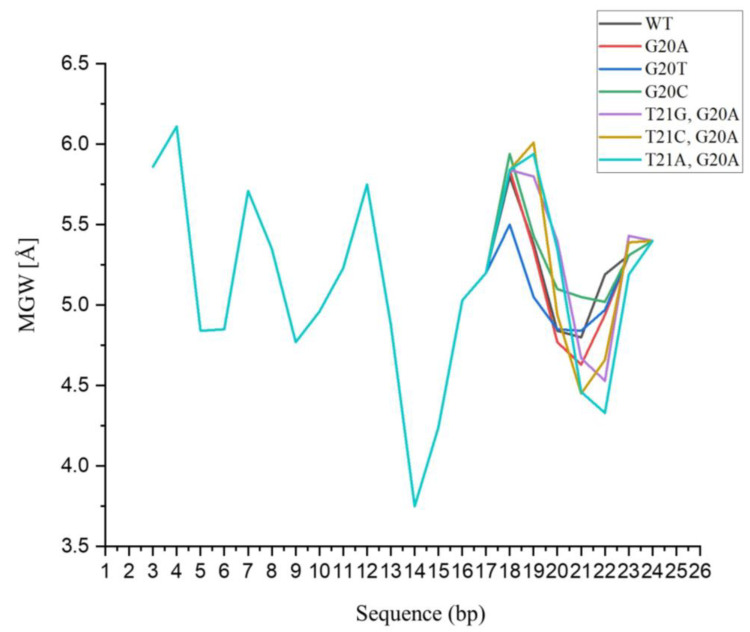
Effect of nucleotide substitution at 20th and 21st positions on MGW of Rv3859c.

**Table 1 microorganisms-11-02505-t001:** Sequences of DNA fragments used in this study.

Fragment Name	Sequence 5′-3′
Rv2524c	5′-GTTTTGTTATCAAATCGTTATGCTGG-3′
Rv3859c Rv0309	5′-GTTACGTGATCGAATTGTGGTCCAGG-3′ 5′-GCTTTGTCACCGCACAGTGACCCAGC-3′
No-Binding	5′-ACAGCGGGACGTTTTCGCGCTGCATC-3′

**Table 2 microorganisms-11-02505-t002:** The accession numbers of the sequences used for alignment.

Species	Accession Number		
	MtrA	PhoP	PhoB
*M. tuberculosis*	P9WGM	P71814	TXA41531
*M. abscessus*	WP_005056284	SHU14306.1	SHP57312.1
*M. smegmatis*	ABK69775.1	WP_252461879.1	WP_228095947.1
*M. marinum*	ACC39757.1	WP_117426730.1	WP_117423619.1
*M. avium*	WP_033725540.1	WP_019730214.1	WP_233082555.1
*M. bovis*	TXA40256.1	TXA39382.1	TXA41531.1
*M. chelonae*	VEG07610.1	MBB4853268.1	MBB4852786.1

**Table 3 microorganisms-11-02505-t003:** Binding affinity distribution of MtrA.

Affinity (KD (M))	Number
High (1 × 10^−7~−8^)	13
Medium (1 × 10^−5~−6^)	72
Low (1 × 10^−3~−4^)	209

**Table 4 microorganisms-11-02505-t004:** The role of the 6–11 and 16–21 regions in Rv2524c affinity.

Fragment	KD (M)	R^2^
Wild type	6.25 × 10^−8^	0.966
6–11 replaced	5.30 × 10^−3^	0.823
16–21 replaced	ND	/

**Table 5 microorganisms-11-02505-t005:** Hydrogen bonds involved in dimer formation.

S. No	MtrA_1_	Domain	Dist. [Å]	MtrA_2_	Domain
1	Arg 121 [ NH2]	REC (αH5)	2.81	Leu 95 [ O]	REC (αH4)
2	Arg 117 [ NH2]	REC (αH5)	2.49	Glu 96 [ OE2]	REC (αH4)
3	Arg 121 [ NH2]	REC (αH5)	3.10	Ala 99 [ O]	REC (Coil-9)
4	Arg 119 [ NH1]	REC (αH5)	2.88	Asp 100 [ O]	REC (Coil-9)
5	Arg 122 [ NH2]	REC (Coil-11)	3.70	Asp 100 [ OD1]	REC (Coil-9)
6	Arg 115 [ NH1]	REC (αH5)	2.91	Asp 101 [ OD1]	REC (β-5)
7	Arg 115 [ NH2]	REC (αH5)	3.04	Tyr 102 [ O]	REC (β-5)
8	Arg 122 [ NH1]	REC (Coil-11)	3.06	Arg 119 [ O]	REC (αH5)
9	Arg 170 [ NH2]	DBD (Coil-15)	2.56	Asp 139 [ OD2]	DBD (Coil-12)
10	Val 172 [ N]	DBD (Coil-15)	3.18	Ala 142 [ O]	DBD (Coil-12)
11	Glu 96 [ O]	REC (αH4)	3.85	Arg 121 [ NH2]	REC (αH5)
12	Glu 96 [ OE1]	REC (αH4)	2.57	Arg 121 [ NH1]	REC (αH5)
13	Glu 96 [ OE2]	REC (αH4)	2.67	Arg 117 [ NE]	REC (αH5)
14	Asp 100 [ O]	REC (Coil-9)	2.88	Arg 119 [ NH1]	REC (αH5)
15	Asp 101 [ OD1]	REC (β-5)	2.85	Arg 115 [ NH1]	REC (αH5)
16	Tyr 102 [ O]	REC (β-5)	3.14	Arg 115 [ NH2]	REC (αH5)
17	Glu 129 [ OE2]	REC (Coil-11)	2.93	Asn 123 [ ND2]	REC (Coil-11)

**Table 6 microorganisms-11-02505-t006:** Effect of Rv3859c 20th- and 21st-position nucleotide substitution on affinities.

Fragment	KD (M)
WT	1.33 × 10^−5^
G20A	1.78 × 10^−7^
G20T	4.38 × 10^−3^
G20C	7.12 × 10^−3^
T21G, G20A	1.57 × 10^−5^
T21C, G20A	1.24 × 10^−7^
T21A, G20A	1. 95 × 10^−5^

## Data Availability

The data that support the findings of this study are available from the corresponding author upon reasonable request.

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
