# Peer review of "Substrate DNA Promoting Binding of Mycobacterium tuberculosis MtrA by Facilitating Dimerization and Interpretation of Affinity by Minor Groove Width"

_microorganisms, 2023, doi:10.3390/microorganisms11102505_

Round 1

Reviewer 1 Report

Reviewer comments

Manuscript: microorganisms-2568010 - Substrate DNA Promoting Binding of Mycobacterium tuberculosis MtrA by facilitating Dimerization and Interpretation of Affinity by MGW.

The authors attempted to explore the mechanism of MtrA response regulator of Mycobacterium tuberculosis specificity and affinity determination comprehensively from sequence to the structure, so as to elucidate how MtrA differentially regulating hundreds of targeted genes. The results of binding box replacement proved that box 2 was essential for binding, which implied the differential roles of two boxes in binding process. The results of substitution of the nucleotide at position 20th and/or 21st indicated that the affinity was negatively associated with the value of MGW precisely at position 21st. The dimerization of unphosphorylated MtrA facilitating by low-affinity DNA was identified for first time. The findings of this study provided new insight into the molecular mechanism of dimerization, binding specificity and affinity determination of MtrA and clues for solving the puzzle of how global TFs regulating large quantity of target genes.

The data analysis methods are correct.

The English of the text is well written and well readable but needs additional checking with a professional translator.

The uniqueness of the text is more than 90% by AntiPlagiarism.NET.

The text contains some misspellings and typos. Also need to expand the part of the discussion.

There are some comments and questions:

Line 61 - for the sentence - MtrA, the only essential response regulator of M. tuberculosis [18] - add additional citation (Singh et al., 2020).

Also add to the References - Singh, K.K.; Athira, P.J.; Bhardwaj, N.; Singh, D.P.; Watson, U.; Saini, D.K. Acetylation of Response Regulator Protein MtrA in M. tuberculosis Regulates Its Repressor Activity. Front Microbiol 2020, 11, 516315. doi: 10.3389/fmicb.2020.516315

Lines 79-81 - for the sentence - After TFs were phosphorylated in vitro before co-crystallization, several studies concluded that phosphorylation on REC of OmpR family member was essential for the dimerization [36,37], meanwhile phosphorylation could considerably increase its binding affinity to target DNA [5,32]. - add information about acetylation and add additional citation (Singh et al., 2020).

Line 156 - Shanghi - should be - Shanghai.

Line 168 - In Table 2 - Smegamatics - should be - smegmatis. 

Line 168 - In Table 2 - The Accession number of the sequences used for alignment in Figure 9. - the authors did not use these accession number of the sequences.

Line 199 - In Figure 1 - add explanation of (a), (b), (c) to the figure caption. Add link to the Figure 1 in the text.

Lines 200, 337 - phylogenic - should be - phylogenetic.

Line 207 - Figure 2 - Phylogeny analysis -  - Phylogeny analysis.

Line 218 - in the Table 4 - 5-11-replaced - should be - 6-11-replaced.

Line 297 - lepore - should be - leprae.

Line 339 - drastical - should be - drastic.

In the work I did not find controls in the experiments. Please explain.

Please improve the manuscript according to the above comments and answer questions.

Minor editing of English language required.

Author Response

Thank you very much for taking the time to review our manuscript. Please find the detailed responses below and the corresponding revisions/corrections highlighted/in track changes in the revised manuscript. 

Reviewer 2 Report

The manuscript describes a study of the affinity of a transcription factor for a number of sequences with the aim of understanding the molecular bases of binding affinity and specificity. One can only guess that the study is interesting and scientific sound because the english form is so poor that the text is simply unreadable in some parts and thus very difficult to understand.

The abstract is very confusing and the aim of the study is not clearly stated since the beginning. The use of undefined abbreviations (such as MGW in the title and abstract) adds confusion for the reader.

Other issues are:

Section 2.4. Describe in detail the technique used for binding affinity assays.

Section 2.5. Describe the meaning of lg calculations.

Section 3.1. Use scientifico notation for binding affinity values with the appropriate unit of measure (e.g., 10-8 M).

Section 3.4. What's the meaning of XX at the end of the first paragraph of page 9. Further, in the text the authors state that only one residue of the dimer model is outside the allowed regions of the Ramachandran plot while it seems that more than one point of the plot is outside. Are the additional points glycine residues?

Section 3.6. For most MGW values there is not a significant difference upon nucleotide substitution. For instance at position 21 before and after substitution values differ by only by 0.17 A. Furthermore it is not clear if the affinity values in this case are calculated or derived from experiments.

The english language is very poor and some parts of the manuscript are almost unreadable and thus very difficult to understand. The english form must be carefully revised, better if by a native english speaker, avoiding grammar errors and improving the structure of the sentences. The attached pdf files highligths in yellow the main parts of the text that need to be revised. 

Author Response

(The authors gave the same response as above.)

Round 2

Reviewer 2 Report

The authors made quite an effort to improve the quality of the manuscript. Therefore in my opinion the manuscript can be published in its present form.